# Understanding and Improving Interpolation in Autoencoders via an Adversarial Regularizer

**David Berthelot**[*]
Google Brain
dberth@google.com

**Colin Raffel**[*]
Google Brain
craffel@gmail.com

**Aurko Roy**
Google Brain
aurkor@google.com

**Ian Goodfellow**
Google Brain
goodfellow@google.com

## Abstract

Autoencoders provide a powerful framework for learning compressed representations by encoding all of the information needed to reconstruct a data point in a latent code. In some cases, autoencoders can "interpolate": By decoding the convex combination of the latent codes for two datapoints, the autoencoder can produce an output which semantically mixes characteristics from the datapoints. In this paper, we propose a regularization procedure which encourages interpolated outputs to appear more realistic by fooling a critic network which has been trained to recover the mixing coefficient from interpolated data. We then develop a simple benchmark task where we can quantitatively measure the extent to which various autoencoders can interpolate and show that our regularizer dramatically improves interpolation in this setting. We also demonstrate empirically that our regularizer produces latent codes which are more effective on downstream tasks, suggesting a possible link between interpolation abilities and learning useful representations.

## 1 Introduction

One goal of unsupervised learning is to uncover the underlying structure of a dataset without using explicit labels. A common architecture used for this purpose is the *autoencoder*, which learns to map datapoints to a latent code from which the data can be recovered with minimal information loss. Typically, the latent code is lower dimensional than the data, which indicates that autoencoders can perform some form of dimensionality reduction. For certain architectures, the latent codes have been shown to disentangle important factors of variation in the dataset which makes such models useful for representation learning (Chen et al., 2016a; Higgins et al., 2017). In the past, they were also used for pre-training other networks by being trained on unlabeled data and then being stacked to initialize a deep network (Bengio et al., 2007; Vincent et al., 2010). More recently, it was shown that imposing a prior on the latent space allows autoencoders to be used for probabilistic or generative modeling (Kingma & Welling, 2014; Rezende et al., 2014; Makhzani et al., 2015).

In some cases, autoencoders have shown the ability to *interpolate*. Specifically, by mixing codes in latent space and decoding the result, the autoencoder can produce a semantically meaningful combination of the corresponding datapoints. Interpolation has been frequently reported as a qualitative experimental result in studies about autoencoders (Dumoulin et al., 2016; Bowman et al., 2015; Roberts et al., 2018; Mescheder et al., 2017; Mathieu et al., 2016; Ha & Eck, 2018) and latent-variable generative models in general (Dinh et al., 2016; Radford et al., 2015; van den Oord et al., 2016). The ability to interpolate can be useful in its own right e.g. for creative applications (Carter & Nielsen, 2017). However, it also indicates that the autoencoder can "extrapolate" beyond the training data and has learned a latent space with a particular structure. Specifically, if interpolating between two points in latent space produces a smooth semantic warping in data space, this suggests that nearby points in latent space are semantically similar. A visualization of this idea is shown in fig. 1, where a smooth

---

[*]Equal contribution.

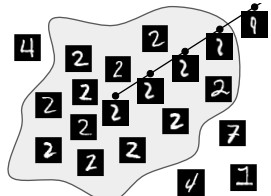

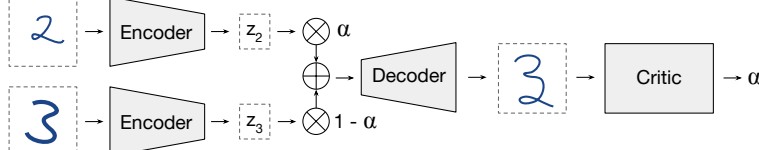

Figure 1: Successful interpolation suggests that semantically similar points may be clustered together in latent space.

Figure 2: Adversarially Constrained Autoencoder Interpolation (ACAI). A critic network is fed interpolants and reconstructions and tries to predict the interpolation coefficient $\alpha$ corresponding to its input (with $\alpha = 0$ for reconstructions). The autoencoder is trained to fool the critic into outputting $\alpha = 0$ for interpolants.

interpolation between a "2" and a "9" suggests that the 2 is surrounded by semantically similar points, i.e. other 2s. This property may suggest that an autoencoder which interpolates well could also provide a good learned representation for downstream tasks because similar points are clustered. If the interpolation is not smooth, there may be "discontinuities" in latent space which could result in the representation being less useful as a learned feature. This connection between interpolation and a "flat" data manifold has been explored in the context of unsupervised representation learning (Bengio et al., 2013b) and regularization (Verma et al., 2018).

Given the widespread use of interpolation as a qualitative measure of autoencoder performance, we believe additional investigation into the connection between interpolation and representation learning is warranted. Our goal in this paper is threefold: First, we introduce a regularization strategy with the specific goal of encouraging improved interpolations in autoencoders (section 2); second, we develop a synthetic benchmark where the slippery concept of a "semantically meaningful interpolation" is quantitatively measurable (section 3.1) and evaluate common autoencoders on this task (section 3.2); and third, we confirm the intuition that good interpolation can result in a useful representation by showing that the improved interpolation ability produced by our regularizer elicits improved representation learning performance on downstream tasks (section 4). We also make our codebase available[1] which provides a unified implementation of many common autoencoders including our proposed regularizer.

## 2 AN ADVERSARIAL REGULARIZER FOR IMPROVING INTERPOLATIONS

Autoencoders, also called auto-associators (Bourlard & Kamp, 1988), consist of the following structure: First, an input $x \in \mathbb{R}^{d_x}$ is passed through an "encoder" $z = f_\theta(x)$ parametrized by $\theta$ to obtain a latent code $z \in \mathbb{R}^{d_z}$. The latent code is then passed through a "decoder" $\hat{x} = g_\phi(z)$ parametrized by $\phi$ to produce an approximate reconstruction $\hat{x} \in \mathbb{R}^{d_x}$ of the input $x$. We consider the case where $f_\theta$ and $g_\phi$ are implemented as multi-layer neural networks. The encoder and decoder are trained simultaneously (i.e. with respect to $\theta$ and $\phi$) to minimize some notion of distance between the input $x$ and the output $\hat{x}$, for example the squared $L_2$ distance $\|x - \hat{x}\|^2$.

Interpolating using an autoencoder describes the process of using the decoder $g_\phi$ to decode a mixture of two latent codes. Typically, the latent codes are combined via a convex combination, so that interpolation amounts to computing $\hat{x}_\alpha = g_\phi(\alpha z_1 + (1-\alpha)z_2)$ for some $\alpha \in [0, 1]$ where $z_1 = f_\theta(x_1)$ and $z_2 = f_\theta(x_2)$ are the latent codes corresponding to data points $x_1$ and $x_2$. We also experimented with spherical interpolation which has been used in settings where the latent codes are expected to have spherical structure (Huszár, 2017; White, 2016; Roberts et al., 2018), but found it made no discernible difference in practice for any autoencoder we studied. Ideally, adjusting $\alpha$ from 0 to 1 will produce a sequence of realistic datapoints where each subsequent $\hat{x}_\alpha$ is progressively less semantically similar to $x_1$ and more semantically similar to $x_2$. The notion of "semantic similarity" is problem-dependent and ill-defined; we discuss this further in section 3.

## 2.1 ADVERSARIALLY CONSTRAINED AUTOENCODER INTERPOLATION (ACAI)

As mentioned above, a high-quality interpolation should have two characteristics: First, that intermediate points along the interpolation are indistinguishable from real data; and second, that the intermediate points provide a semantically smooth morphing between the endpoints. The latter characteristic is hard to enforce because it requires defining a notion of semantic similarity for a given dataset, which is often hard to explicitly codify. So instead, we propose a regularizer which encourages interpolated datapoints to appear realistic, or more specifically, to appear indistinguishable from reconstructions of real datapoints. We find empirically that this constraint results in realistic and smooth interpolations in practice (section 3.1) in addition to providing improved performance on downstream tasks (section 4).

To enforce this constraint we introduce a critic network (Goodfellow et al., 2014) which is fed interpolations of existing datapoints (i.e. $\hat{x}_\alpha$ as defined above). Its goal is to predict $\alpha$ from $\hat{x}_\alpha$, i.e. to predict the mixing coefficient used to generate its input. When training the model, for each pair of training data points we randomly sample a value of $\alpha$ to produce $\hat{x}_\alpha$. In order to resolve the ambiguity between predicting $\alpha$ and $1 - \alpha$, we constrain $\alpha$ to the range $[0, 0.5]$ when feeding $\hat{x}_\alpha$ to the critic. In contrast, the autoencoder is trained to fool the critic to think that $\alpha$ is always zero. This is achieved by adding an additional term to the autoencoder's loss to optimize its parameters to fool the critic. In a loose sense, the critic can be seen as approximating an "adversarial divergence" (Liu et al., 2017; Arora et al., 2017) between reconstructions and interpolants which the autoencoder tries to minimize.

Formally, let $d_\omega(x)$ be the critic network, which for a given input produces a scalar value. The critic is trained to minimize

$$\mathcal{L}_d = \|d_\omega(\hat{x}_\alpha) - \alpha\|^2 + \|d_\omega(\gamma x + (1 - \gamma)\hat{x})\|^2 \tag{1}$$

where, as above, $\hat{x}_\alpha = g_\phi(\alpha f_\theta(x_1) + (1 - \alpha)f_\theta(x_2))$, $\hat{x} = g_\phi(f_\theta(x))$ for some $x$ (not necessarily $x_1$ or $x_2$), and $\gamma$ is a scalar hyperparameter. The first term trains the critic to recover $\alpha$ from $\hat{x}_\alpha$. The second term serves as a regularizer with two functions: First, it enforces that the critic consistently outputs 0 for non-interpolated inputs; and second, by interpolating between $x$ and $\hat{x}$ (the autoencoder's reconstruction of $x$) in data space it ensures the critic is exposed to realistic data even when the autoencoder's reconstructions are poor. We found the second term was not crucial for our approach, but helped stabilize the convergence of the autoencoder and allowed us to use consistent hyperparameters and architectures across all datasets and experiments. The autoencoder's loss function is modified by adding a regularization term:

$$\mathcal{L}_{f,g} = \|x - g_\phi(f_\theta(x))\|^2 + \lambda\|d_\omega(\hat{x}_\alpha)\|^2 \tag{2}$$

where $\lambda$ is a scalar hyperparameter which controls the weight of the regularization term. Note that the regularization term is effectively trying to make the critic output 0 regardless of the value of $\alpha$, thereby "fooling" the critic into thinking that an interpolated input is non-interpolated (i.e., having $\alpha = 0$). The parameters $\theta$ and $\phi$ are optimized with respect to $\mathcal{L}_{f,g}$ (which gives the autoencoder access to the critic's gradients) and $\omega$ is optimized with respect to $\mathcal{L}_d$. We refer to the use of this regularizer as **Adversarially Constrained Autoencoder Interpolation** (ACAI). A diagram of the ACAI is shown in fig. 2. Assuming an effective critic, the autoencoder successfully "wins" this adversarial game by producing interpolated points which are indistinguishable from reconstructed data. We find in practice that encouraging this behavior also produces semantically smooth interpolations and improved representation learning performance, which we demonstrate in the following sections. Our loss function is similar to the one used in the Least Squares Generative Adversarial Network (Mao et al., 2017) in the sense that they both measure the distance between a critic's output and a scalar using a squared L2 loss. However, they are substantially different in that ours is used as a regularizer for autoencoders rather than for generative modeling and our critic attempts to regress the interpolation coefficient $\alpha$ instead of a fixed scalar hyperparameter.

Note that the only thing ACAI encourages is that interpolated points appear realistic. The critic only ever sees a single reconstruction or interpolant at a time; it is never fed real datapoints or latent vectors. It therefore will only be able to successfully recover $\alpha$ if the quality of the autoencoder's output degrades consistently across an interpolation as a function of $\alpha$ (as seen, for example, in fig. 3a where interpolated points become successively blurrier and darker). ACAI's primary purpose is to discourage this behavior. In doing so, it may implicitly modify the structure of the latent space

---

[1]https://github.com/anonymous-iclr-2019/acai-iclr-2019

learned by the autoencoder, but ACAI itself does not directly impose a specific structure. Our goal in introducing ACAI is to test whether simply encouraging better interpolation behavior produces a better representation for downstream tasks. Further, in contrast with the standard Generative Adversarial Network (GAN) setup (Goodfellow et al., 2014), ACAI does not distinguish between "real" and "fake" data; rather, it simply attempts to regress the interpolation coefficient $\alpha$. Furthermore, GANs are a generative modeling technique, not a representation learning technique; in this paper, we focus on autoencoders and their ability to learn useful representations.

# 3   AUTOENCODERS, AND HOW THEY INTERPOLATE

How can we measure whether an autoencoder interpolates effectively and whether our proposed regularization strategy achieves its stated goal? As mentioned in section 2, defining interpolation relies on the notion of "semantic similarity" which is a vague and problem-dependent concept. For example, a definition of interpolation along the lines of "$\alpha z_1 + (1 - \alpha)z_2$ should map to $\alpha x_1 + (1 - \alpha)x_2$" is overly simplistic because interpolating in "data space" often does not result in realistic datapoints – in images, this corresponds to simply fading between the pixel values of the two images. Instead, we might hope that our autoencoder smoothly morphs between salient characteristics of $x_1$ and $x_2$, even when they are dissimilar. Put another way, we might hope that decoded points along the interpolation smoothly traverse the underlying manifold of the data instead of simply interpolating in data space. However, we rarely have access to the underlying data manifold. To make this problem more concrete, we introduce a simple benchmark task where the data manifold is simple and known a priori which makes it possible to quantify interpolation quality. We then evaluate the ability of various common autoencoders to interpolate on our benchmark. Finally, we test ACAI on our benchmark and show that it exhibits dramatically improved performance and qualitatively superior interpolations.

## 3.1   AUTOENCODING LINES

Given that the concept of interpolation is difficult to pin down, our goal is to define a task where a "correct" interpolation between two datapoints is unambiguous and well-defined. This will allow us to quantitatively evaluate the extent to which different autoencoders can successfully interpolate. Towards this goal, we propose the task of autoencoding $32 \times 32$ grayscale images of lines. We consider 16-pixel-long lines beginning from the center of the image and extending outward at an angle $\Lambda \in [0, 2\pi]$ (or put another way, lines are radii of the circle circumscribed within the image borders). An example of 16 such images is shown in fig. 4a (appendix A.1). In this task, the data manifold can be defined entirely by a single variable: $\Lambda$. We can therefore define a valid interpolation from $x_1$ to $x_2$ as one which smoothly and linearly adjusts $\Lambda$ from the angle of the line in $x_1$ to the angle in $x_2$. We further require that the interpolation traverses the shortest path possible along the data manifold. We provide some concrete examples of good and bad interpolations, shown and described in appendix A.1.

On any dataset, our desiderata for a successful interpolation are that intermediate points look realistic and provide a semantically meaningful morphing between its endpoints. On this synthetic lines dataset, we can formalize these notions as specific evaluation metrics, which we describe in detail in appendix A.2. To summarize, we propose two metrics: Mean Distance and Smoothness. Mean Distance measures the average distance between interpolated points and "real" datapoints. Smoothness measures whether the angles of the interpolated lines follow a linear trajectory between the angle of the start and endpoint. Both of these metrics are simple to define due to our construction of a dataset where we exactly know the data distribution and manifold; we provide a full definition and justification in appendix A.2. A perfect alignment would achieve 0 for both scores; larger values indicate a failure to generate realistic interpolated points or produce a smooth interpolation respectively. By choosing a synthetic benchmark where we can explicitly measure the quality of an interpolation, we can confidently evaluate different autoencoders on their interpolation abilities.

To evaluate an autoencoder on the synthetic lines task, we randomly sample line images during training and compute our evaluation metrics on a separate randomly-sampled test set of images. Note that we never train any autoencoder explicitly to produce an optimal interpolation; "good" interpolation is an emergent property which occurs only when the architecture, loss function, training procedure, etc. produce a suitable latent space.

Table 1: Scores achieved by different autoencoders on the synthetic line benchmark (lower is better).

| Metric | Baseline | Denoising | VAE | AAE | VQ-VAE | ACAI |
|---|---|---|---|---|---|---|
| **Mean Distance** $(\times 10^{-3})$ | 6.88±0.21 | 4.21±0.32 | 1.21±0.17 | 3.26±0.19 | 5.41±0.49 | **0.24±0.01** |
| **Smoothness** | 0.44±0.04 | 0.66±0.02 | 0.49±0.13 | 0.14±0.02 | 0.77±0.02 | **0.10±0.01** |

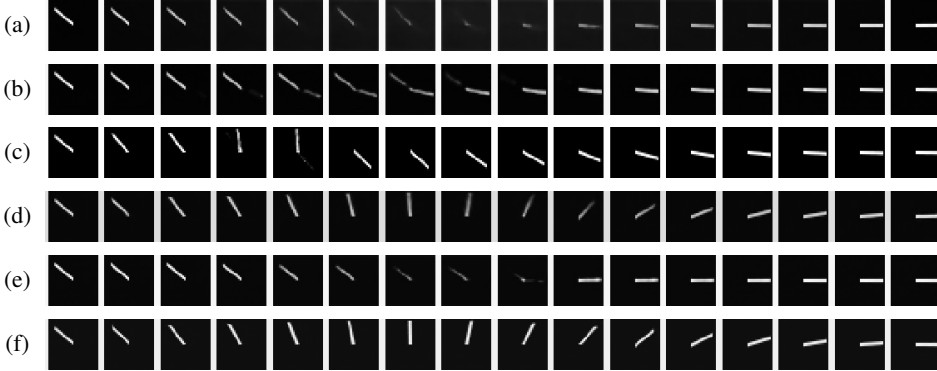

Figure 3: Interpolations on the synthetic lines benchmark produced by (a) baseline auto-encoder, (b) denoising autoencoder, (c) Variational Autoencoder, (d) Adversarial Autoencoder, (e) Vector Quantized Variational Autoencoder, (f) Adversarially Constrained Autoencoder Interpolation (our model). While we only show one example here, the behavior of each autoencoder was generally similar for all interpolations. A more comprehensive measure of interpolation behavior is given in table 1.

## 3.2 AUTOENCODERS

In this section, we describe various common autoencoder structures and objectives and try them on the lines task. Our goal is to quantitatively evaluate the extent to which standard autoencoders exhibit useful interpolation behavior. Our results, which we describe below, are summarized in table 1.

**Base Model** Perhaps the most basic autoencoder structure is one which simply maps input data-points through a "bottleneck" layer whose dimensionality is smaller than the input. In this setup, $f_\theta$ and $g_\phi$ are both neural networks which respectively map the input to a deterministic latent code $z$ and then back to a reconstructed input. Typically, $f_\theta$ and $g_\phi$ are trained simultaneously with respect to $\|x - \hat{x}\|^2$. We will use this framework as a baseline for experimentation for all of the autoencoder variants discussed below. In particular, for our base model and all of the other autoencoders we will use the model architecture and training procedure described in appendix B. As a short summary, our encoder consists of a stack of convolutional and average pooling layers, whereas the decoder consists of convolutional and nearest-neighbor upsampling layers. For experiments on the synthetic "lines" task, we use a latent dimensionality of $64$. Note that, because the data manifold is effectively one-dimensional, we might expect autoencoders to be able to model this dataset using a one-dimensional latent code; however, using a larger latent code reflects the realistic scenario where the latent space is larger than necessary. After training our baseline autoencoder, we achieved a Mean Distance score which was the worst (highest) of all of the autoencoders we studied, though the Smoothness was on par with various other approaches. In general, we observed some reasonable interpolations when using the baseline model, but found that the intermediate points on the interpolation were typically not realistic as seen in the example interpolation in fig. 3a.

**Denoising Autoencoder** An early modification to the standard autoencoder setup was proposed by Vincent et al. (2010), where instead of feeding $x$ into the autoencoder, a corrupted version $\tilde{x} \sim q(\tilde{x}|x)$ is sampled from the conditional probability distribution $q(\tilde{x}|x)$ and is fed into the autoencoder instead. The autoencoder's goal remains to produce $\hat{x}$ which minimizes $\|x - \hat{x}\|^2$. One justification of this approach is that the corrupted inputs should fall outside of the true data manifold, so the autoencoder must learn to map points from outside of the data manifold back onto it. This provides an implicit way of defining and learning the data manifold via the coordinate system induced by the latent space.

While various corruption procedures $q(\tilde{x}|x)$ have been used such as masking and salt-and-pepper noise, in this paper we consider the simple case of additive isotropic Gaussian noise where $\tilde{x} \sim \mathcal{N}(x, \sigma^2 I)$ and $\sigma$ is a hyperparameter. After tuning $\sigma$, we found simply setting $\sigma = 1.0$ to work best. Interestingly, we found the denoising autoencoder often produced "data-space" interpolation (as seen in fig. 3b) when interpolating in latent space. This resulted in comparatively poor Mean Distance and Smoothness scores.

**Variational Autoencoder**    The Variational Autoencoder (VAE) (Kingma & Welling, 2014; Rezende et al., 2014) introduces the constraint that the latent code $z$ is a random variable distributed according to a prior distribution $p(z)$. The encoder $f_\theta$ can then be considered an approximation to the posterior $p(z|x)$. Then, the decoder $g_\phi$ is taken to parametrize the likelihood $p(x|z)$; in all of our experiments, we consider $x$ to be Bernoulli distributed. The latent distribution constraint is enforced by an additional loss term which measures the KL divergence between approximate posterior and prior. VAEs then use log-likelihood for the reconstruction loss (cross-entropy in the case of Bernoulli-distributed data), which results in the following combined loss function: $-\mathbb{E}[\log g_\phi(z)] + \text{KL}(f_\theta(x)||p(z))$ where the expectation is taken with respect to $z \sim f_\theta(x)$ and $\text{KL}(\cdot||\cdot)$ is the KL divergence. Minimizing this loss function can be considered maximizing a lower bound (the "ELBO") on the likelihood of the training set, producing a likelihood-based generative model which allows novel data points to be sampled by first sampling $z \sim p(z)$ and then computing $g_\phi(z)$. A common choice is to let $q(z|x)$ be a diagonal-covariance Gaussian, in which case backpropagation through sampling from $q(z|x)$ is feasible via the "reparametrization trick" which replaces $z \sim \mathcal{N}(\mu, \sigma I)$ with $\epsilon \sim \mathcal{N}(0, I), z = \mu + \sigma \odot \epsilon$ where $\mu, \sigma \in \mathbb{R}^{d_z}$ are the predicted mean and standard deviation produced by $f_\theta$. Various modified objectives (Higgins et al., 2017; Zhao et al., 2017), improved prior distributions (Kingma et al., 2016; Tomczak & Welling, 2016; 2017) and improved model architectures (Sønderby et al., 2016; Chen et al., 2016b; Gulrajani et al., 2016) have been proposed to better the VAE's performance on downstream tasks, but in this paper we solely consider the "vanilla" VAE objective and prior described above applied to our baseline autoencoder structure.

When trained on the lines benchmark, we found the VAE was able to effectively model the data distribution (see samples, fig. 5 in appendix C) and accurately reconstruct inputs. In interpolations produced by the VAE, intermediate points tend to look realistic, but the angle of the lines do not follow a smooth or short path (fig. 3c). This resulted in a very good Mean Distance score but a very poor Smoothness score. Contrary to expectations, this suggests that *desirable interpolation behavior may not follow from an effective generative model of the data distribution.*

**Adversarial Autoencoder**    The Adversarial Autoencoder (AAE) (Makhzani et al., 2015) proposes an alternative way of enforcing structure on the latent code. Instead of minimizing a KL divergence between the distribution of latent codes and a prior distribution, a critic network is trained in tandem with the autoencoder to predict whether a latent code comes from $f_\theta$ or from the prior $p(z)$. The autoencoder is simultaneously trained to reconstruct inputs (via a standard reconstruction loss) and to "fool" the critic. The autoencoder is allowed to backpropagate gradients through the critic's loss function, but the autoencoder and critic parameters are optimized separately. This effectively computes an "adversarial divergence" between the latent code distribution and the chosen prior. This framework was later generalized and referred to as the "Wasserstein Autoencoder" (Tolstikhin et al., 2017) One advantage of this approach is that it allows for an arbitrary prior (as opposed to those which have a tractable KL divergence). The disadvantages are that the AAE no longer has a probabilistic interpretation and involves optimizing a minimax game, which can cause instabilities.

Using the AAE requires choosing a prior, a critic structure, and a training scheme for the critic. For simplicity, we also used a spherical Gaussian prior for the AAE. We experimented with various architectures for the critic, and found the best performance with a critic which consisted of two dense layers, each with 100 units and a leaky ReLU nonlinearity. We found it satisfactory to simply use the same optimizer and learning rate for the critic as was used for the autoencoder. On our lines benchmark, the AAE typically produced smooth interpolations, but exhibited degraded quality in the middle of interpolations (fig. 3d). This behavior produced the best Smoothness score among existing autoencoders, but a relatively poor Mean Distance score.

**Vector Quantized Variational Autoencoder (VQ-VAE)**    The Vector Quantized Variational Autoencoder (VQ-VAE) was introduced by (van den Oord et al., 2017) as a way to train discrete-latent

autoencoders using a learned codebook. In the VQ-VAE, the encoder $f_\theta(x)$ produces a continuous hidden representation $z \in \mathbb{R}_z^d$ which is then mapped to $z_q$, its nearest neighbor in a "codebook" $\{e_j \in \mathbb{R}^{d_z}, j \in 1, \ldots, K\}$. $z_q$ is then passed to the decoder for reconstruction. The encoder is trained to minimize the reconstruction loss using the straight-through gradient estimator (Bengio et al., 2013a), together with a *commitment loss* term $\beta \|z - \text{sg}(z_q)\|$ (where $\beta$ is a scalar hyperparameter) which encourages encoder outputs to move closer to their nearest codebook entry. Here sg denotes the stop gradient operator, i.e. $\text{sg}(x) = x$ in the forward pass, and $\text{sg}(x) = 0$ in the backward pass. The codebook entries $e_j$ are updated as an exponential moving average (EMA) of the continuous latents $z$ that map to them at each training iteration. The VQ-VAE training procedure using this EMA update rule can be seen as performing the $K$-means or the hard Expectation Maximization (EM) algorithm on the latent codes (Roy et al., 2018).

We perform interpolation in the VQ-VAE by interpolating continuous latents, mapping them to their nearest codebook entries, and decoding the result. Assuming sufficiently large codebook, a semantically "smooth" interpolation may be possible. On the lines task, we found that this procedure produced poor interpolations. Ultimately, many entries of the codebook were mapped to unrealistic datapoints, and the interpolations resembled those of the baseline autoencoder.

**Adversarially Constrained Autoencoder Interpolation**    Finally, we turn to evaluating our proposed adversarial regularizer for improving interpolations. For simplicity, on the lines benchmark we found it sufficient to use a critic architecture which was equivalent to the encoder (as described in appendix B). To produce a single scalar value from its output, we computed the mean of its final layer activations. For the hyperparameters $\lambda$ and $\gamma$ we found values of $0.5$ and $0.2$ to achieve good results, though the performance was not very sensitive to their values. We use these values for the coefficients for all of our experiments. Finally, we trained the critic using the same optimizer and hyperparameters as the autoencoder.

We found dramatically improved performance on the lines benchmark when using ACAI – it achieved the best Mean Distance and Smoothness score among the autoencoders we considered. When inspecting the resulting interpolations, we found it occasionally chose a longer path than necessary but typically produced "perfect" interpolation behavior as seen in fig. 3f. This provides quantitative evidence ACAI is successful at encouraging realistic and smooth interpolations.

### 3.3    Interpolations on Real Data

We have so far only discussed results on our synthetic lines benchmark. We also provide example reconstructions and interpolations produced by each autoencoder for MNIST (LeCun, 1998), SVHN (Netzer et al., 2011), and CelebA (Liu et al., 2015) in appendix D. For each dataset, we trained autoencoders with latent dimensionalities of 32 and 256. Since we do not know the underlying data manifold for these datasets, no metrics are available to evaluate performance and we can only make qualitative judgments as to the reconstruction and interpolation quality. We find that most autoencoders produce "blurrier" images with $d_z = 32$ but generally give smooth interpolations regardless of the latent dimensionality. The exception to this observation was the VQ-VAE which seems generally to work *better* with $d_z = 32$ and occasionally even diverged for $d_z = 256$ (see e.g. fig. 9e). This may be due to the nearest-neighbor discretization (and gradient estimator) failing in high dimensions. Across datasets, we found the VAE and denoising autoencoder typically produced more blurry interpolations. AAE and ACAI generally produced realistic interpolations, even between dissimilar datapoints (for example, in fig. 7 bottom). The baseline model often effectively interpolated in data space.

### 4    Improved Representation Learning

We have so far solely focused on measuring the interpolation abilities of different autoencoders. Now, we turn to the question of whether improved interpolation is associated with improved performance on downstream tasks. Specifically, we will evaluate whether using our proposed regularizer results in latent space representations which provide better performance in supervised learning and clustering. Put another way, we seek to test whether improving interpolation results in a latent representation which has disentangled important factors of variation (such as class identity) in the dataset. To answer this question, we ran classification and clustering experiments using the learned latent spaces of

Table 2: Single-layer classifier accuracy achieved by different autoencoders.

| Dataset | $d_z$ | Baseline | Denoising | VAE | AAE | VQ-VAE | ACAI |
|---------|-------|----------|-----------|-----|-----|--------|------|
| MNIST | 32 | 94.90±0.14 | 96.00±0.27 | 96.56±0.31 | 70.74±3.27 | 97.50±0.18 | **98.25±0.11** |
| | 256 | 93.94±0.13 | 98.51±0.04 | 98.74±0.14 | 90.03±0.54 | 97.25±1.42 | **99.00±0.08** |
| SVHN | 32 | 26.21±0.42 | 25.15±0.78 | 29.58±3.22 | 23.43±0.79 | 24.53±1.33 | **34.47±1.14** |
| | 256 | 22.74±0.05 | 77.89±0.35 | 66.30±1.06 | 22.81±0.24 | 44.94±20.42 | **85.14±0.20** |
| CIFAR-10 | 256 | 47.92±0.20 | **53.78±0.36** | 47.49±0.22 | 40.65±1.45 | 42.80±0.44 | 52.77±0.45 |
| | 1024 | 51.62±0.25 | 60.65±0.14 | 51.39±0.46 | 42.86±0.88 | 16.22±12.44 | **63.99±0.47** |

Table 3: Clustering accuracy for using K-Means on the latent space of different autoencoders (left) and previously reported methods (right). On the right, "Data" refers to performing K-Means directly on the data and DEC, RIM, and IMSAT are the methods proposed in (Xie et al., 2016; Krause et al., 2010; Hu et al., 2017) respectively. Results marked * are excerpted from (Hu et al., 2017) and ** are from (Xie et al., 2016).

| Dataset | $d_z$ | Baseline | Denoising | VAE | AAE | VQ-VAE | ACAI | Data | DEC | RIM | IMSAT |
|---------|-------|----------|-----------|-----|-----|--------|------|------|-----|-----|-------|
| MNIST | 32 | 77.56 | 82.59 | 75.74 | 79.19 | 82.39 | **94.38** | 53.2* | 84.3** | 58.5* | 98.4* |
| | 256 | 53.70 | 70.89 | 83.44 | 81.00 | **96.80** | 96.17 | | | | |
| SVHN | 32 | 19.38 | 17.91 | 16.83 | 17.35 | 15.19 | **20.86** | 17.9* | 11.9* | 26.8* | 57.3* |
| | 256 | 15.62 | **31.49** | 11.36 | 13.59 | 18.84 | 24.98 | | | | |

different autoencoders on the MNIST (LeCun, 1998), SVHN (Netzer et al., 2011), and CIFAR-10 (Krizhevsky, 2009) datasets.

**Single-Layer Classifier** A common method for evaluating the quality of a learned representation (such as the latent space of an autoencoder) is to use it as a feature representation for a simple, one-layer classifier (i.e. logistic regression) trained on a supervised learning task (Coates et al., 2011). The justification for this evaluation procedure is that a learned representation which has effectively disentangled class identity will allow the classifier to obtain reasonable performance despite its simplicity. To test different autoencoders in this setting, we trained a separate single-layer classifier in tandem with the autoencoder using the latent representation as input. We did not optimize autoencoder parameters with respect to the classifier's loss, which ensures that we are measuring unsupervised representation learning performance. We repeated this procedure for latent dimensionalities of 32 and 256 (MNIST and SVHN) and 256 and 1024 (CIFAR-10).

Our results are shown in table 2. In all settings, using ACAI instead of the baseline autoencoder upon which it is based produced significant gains – most notably, on SVHN with a latent dimensionality of 256, the baseline achieved an accuracy of only 22.74% whereas ACAI achieved 85.14%. In general, we found the denoising autoencoder, VAE, and ACAI obtained significantly higher performance compared to the remaining models. On MNIST and SVHN, ACAI achieved the best accuracy by a significant margin; on CIFAR-10, the performance of ACAI and the denoising autoencoder was similar. By way of comparison, we found a single-layer classifier applied directly to (flattened) image pixels achieved an accuracy of 92.31%, 23.48%, and 39.70% on MNIST, SVHN, and CIFAR-10 respectively, so classifying using the representation learned by ACAI provides a huge benefit.

**Clustering** If an autoencoder groups points with common salient characteristics close together in latent space without observing any labels, it arguably has uncovered some important structure in the data in an unsupervised fashion. A more difficult test of an autoencoder is therefore clustering its latent space, i.e. separating the latent codes for a dataset into distinct groups without using any labels. To test the clusterability of the latent spaces learned by different autoencoders, we simply apply K-Means clustering (MacQueen, 1967) to the latent codes for a given dataset. Since K-Means uses Euclidean distance, it is sensitive to each dimension's relative variance. We therefore used PCA whitening on the latent space learned by each autoencoder to normalize the variance of its dimensions prior to clustering. K-Means can exhibit highly variable results depending on how it is initialized, so for each autoencoder we ran K-Means 1,000 times from different random initializations and chose the clustering with the best objective value on the training set. For evaluation, we adopt the methodology of Xie et al. (2016); Hu et al. (2017): Given that the dataset in question has labels (which are not used

for training the model, the clustering algorithm, or choice of random initialization), we can cluster the data into $C$ distinct groups where $C$ is the number of classes in the dataset. We then compute the "clustering accuracy", which is simply the accuracy corresponding to the optimal one-to-one mapping of cluster IDs and classes (Xie et al., 2016).

Our results are shown in table 3. On both MNIST and SVHN, ACAI achieved the best or second-best performance for both $d_z = 32$ and $d_z = 256$. We do not report results on CIFAR-10 because all of the autoencoders we studied achieved a near-random clustering accuracy. Previous efforts to evaluate clustering performance on CIFAR-10 use learned feature representations from a convolutional network trained on ImageNet (Hu et al., 2017) which we believe only indirectly measures unsupervised learning capabilities.

## 5 CONCLUSION

In this paper, we have provided an in-depth perspective on interpolation in autoencoders. We proposed Adversarially Constrained Autoencoder Interpolation (ACAI), which uses a critic to encourage interpolated datapoints to be more realistic. To make interpolation a quantifiable concept, we proposed a synthetic benchmark and showed that ACAI substantially outperformed common autoencoder models. This task also yielded unexpected insights, such as that a VAE which has effectively learned the data distribution might not interpolate. We also studied the effect of improved interpolation on downstream tasks, and showed that ACAI led to improved performance for feature learning and unsupervised clustering. These findings confirm our intuition that improving the interpolation abilities of a baseline autoencoder can also produce a better learned representation for downstream tasks. However, we emphasize that we do not claim that good interpolation always implies a good representation – for example, the AAE produced smooth and realistic interpolations but fared poorly in our representations learning experiments and the denoising autoencoder had low-quality interpolations but provided a useful representation.

In future work, we are interested in investigating whether our regularizer improves the performance of autoencoders other than the standard "vanilla" autoencoder we applied it to. In this paper, we primarily focused on image datasets due to the ease of visualizing interpolations, but we are also interested in applying these ideas to non-image datasets.

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

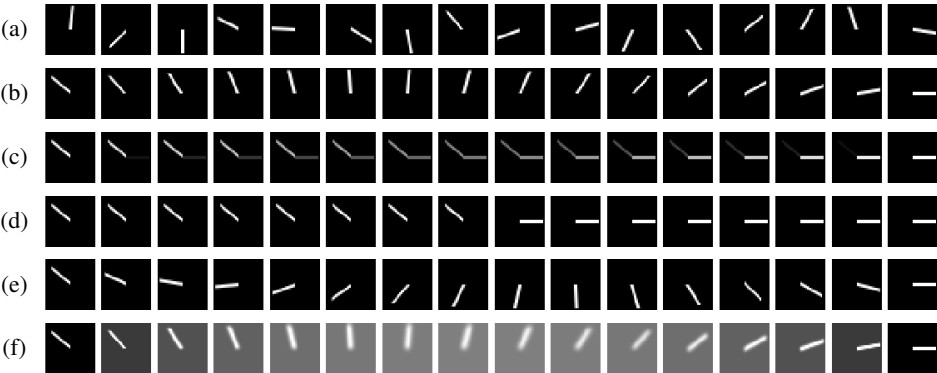

Figure 4: Examples of data and interpolations from our synthetic lines dataset. (a) 16 random samples from the dataset. (b) A perfect interpolation from $\Lambda = {}^{11\pi}/{}_{14}$ to 0. (c) Interpolating in data space rather than "semantic" or latent space. Clearly, interpolating in this way produces points not on the data manifold. (d) An interpolation which abruptly changes from one image to the other, rather than smoothly changing. (e) A smooth interpolation which takes a longer path from the start to end point than necessary. (f) An interpolation which takes the correct path but where intermediate points are not realistic.

## A   LINE BENCHMARK

### A.1   EXAMPLE INTERPOLATIONS

Some example data and interpolations for our synthetic lines benchmark are shown in fig. 4. Full discussion of this benchmark is available in section 3.1.

### A.2   EVALUATION METRICS

We define our Mean Distance and Smoothness metrics as follows: Let $x_1$ and $x_2$ be two input images we are interpolating between and

$$\hat{x}_n = g_\phi \left( \frac{n-1}{N-1} z_1 + \left( 1 - \frac{n-1}{N-1} \right) z_2 \right) \tag{3}$$

be the decoded point corresponding to mixing $x_1$ and $x_2$'s latent codes using coefficient $\alpha = {}^{n-1}/{}_{N-1}$. The images $\hat{x}_n, n \in \{1, \ldots, N\}$ then comprise a length-$N$ interpolation between $x_1$ and $x_2$. To produce our evaluation metrics, we first find the closest true datapoint (according to cosine distance) for each of the $N$ intermediate images along the interpolation. Finding the closest image among all possible line images is infeasible; instead we first generate a size-$D$ collection of line images $\mathcal{D}$ with corresponding angles $\Lambda_q, q \in \{1, \ldots, D\}$ spaced evenly between 0 and $2\pi$. Then, we match each image in the interpolation to a real datapoint by finding

$$C_{n,q} = 1 - \frac{\hat{x}_n \mathcal{D}_q}{\|\hat{x}_n\| \|\mathcal{D}_q\|} \tag{4}$$

$$q_n^\star = \arg\min_q C_{n,q} \tag{5}$$

for $n \in \{1, \ldots, N\}$, where $C_{n,q}$ is the cosine distance between $\hat{x}_n$ and the $q$th entry of $\mathcal{D}$. To capture the notion of "intermediate points look realistic", we compute

$$\text{Mean Distance}(\{\hat{x}_1, \hat{x}_2, \ldots, \hat{x}_N\}) = \frac{1}{N} \sum_{n=1}^{N} C_{n,q_n^\star} \tag{6}$$

We now define a perfectly smooth interpolation to be one which consists of lines with angles which linearly move from the angle of $\mathcal{D}_{q_1^\star}$ to that of $\mathcal{D}_{q_N^\star}$. Note that if, for example, the interpolated lines go from $\Lambda_{q_1^\star} = \pi/10$ to $\Lambda_{q_N^\star} = {}^{19\pi}/{}_{10}$ then the angles corresponding to the shortest path will

have a discontinuity from $0$ to $2\pi$. To avoid this, we first "unwrap" the angles $\{\Lambda_{q_1^\star}, \ldots, \Lambda_{q_N^\star}\}$ by removing discontinuities larger than $\pi$ by adding multiples of $\pm 2\pi$ when the absolute difference between $\Lambda_{q_{n-1}^\star}$ and $\Lambda_{q_n^\star}$ is greater than $\pi$ to produce the angle sequence $\{\tilde{\Lambda}_{q_1^\star}, \ldots, \tilde{\Lambda}_{q_N^\star}\}$.[2] Then, we define a measure of smoothness as

$$\text{Smoothness}(\{\hat{x}_1, \hat{x}_2, \ldots, \hat{x}_N\}) = \frac{1}{|\tilde{\Lambda}_{q_1^\star} - \tilde{\Lambda}_{q_N^\star}|} \max_{n \in \{1, \ldots, N-1\}} \left( \tilde{\Lambda}_{q_{n+1}^\star} - \tilde{\Lambda}_{q_n^\star} \right) - \frac{1}{N-1} \quad (7)$$

In other words, we measure the how much larger the largest change in (normalized) angle is compared to the minimum possible value ($1/(N-1)$).

By way of example, figs. 4b, 4d and 4e would all achieve Mean Distance scores near zero and figs. 4c and 4f would achieve larger Mean Distance scores. Figures 4b and 4f would achieve Smoothness scores near zero, figs. 4c and 4d have poor Smoothness, and fig. 4e is in between.

## B    BASE MODEL ARCHITECTURE AND TRAINING PROCEDURE

All of the autoencoder models we studied in this paper used the following architecture and training procedure: The encoder consists of blocks of two consecutive $3 \times 3$ convolutional layers followed by $2 \times 2$ average pooling. All convolutions (in the encoder and decoder) are zero-padded so that the input and output height and width are equal. The number of channels is doubled before each average pooling layer. Two more $3 \times 3$ convolutions are then performed, the last one without activation and the final output is used as the latent representation. All convolutional layers except for the final use a leaky ReLU nonlinearity (Maas et al., 2013). For experiments on the synthetic "lines" task, the convolution-average pool blocks are repeated 4 times until we reach a latent dimensionality of 64. For subsequent experiments on real datasets (section 4), we repeat the blocks 3 times, resulting in a latent dimensionality of 256.

The decoder consists of blocks of two consecutive $3 \times 3$ convolutional layers with leaky ReLU nonlinearities followed by $2 \times 2$ nearest neighbor upsampling (Odena et al., 2016). The number of channels is halved after each upsampling layer. These blocks are repeated until we reach the target resolution ($32 \times 32$ in all experiments). Two more $3 \times 3$ convolutions are then performed, the last one without activation and with a number of channels equal to the number of desired colors.

All parameters are initialized as zero-mean Gaussian random variables with a standard deviation of $1/\sqrt{\texttt{fan\_in}(1+0.2^2)}$ set in accordance with the leaky ReLU slope of $0.2$. Models are trained on $2^{24}$ samples in batches of size $64$. Parameters are optimized with Adam (Kingma & Welling, 2014) with a learning rate of $0.0001$ and default values for $\beta_1$, $\beta_2$, and $\epsilon$.

## C    VAE SAMPLES ON THE LINE BENCHMARK

In fig. 5, we show some samples from our VAE trained on the synthetic line benchmark. The VAE generally produces realistic samples and seems to cover the data distribution well, despite the fact that it does not produce high-quality interpolations (fig. 3c).

## D    INTERPOLATION EXAMPLES ON REAL DATA

In this section, we provide a series of figures (figs. 6 to 11) showing interpolation behavior for the different autoencoders we studied. Further discussion of these results is available in section 3.3

---

[2]See    e.g.    https://docs.scipy.org/doc/numpy/reference/generated/numpy.unwrap.html

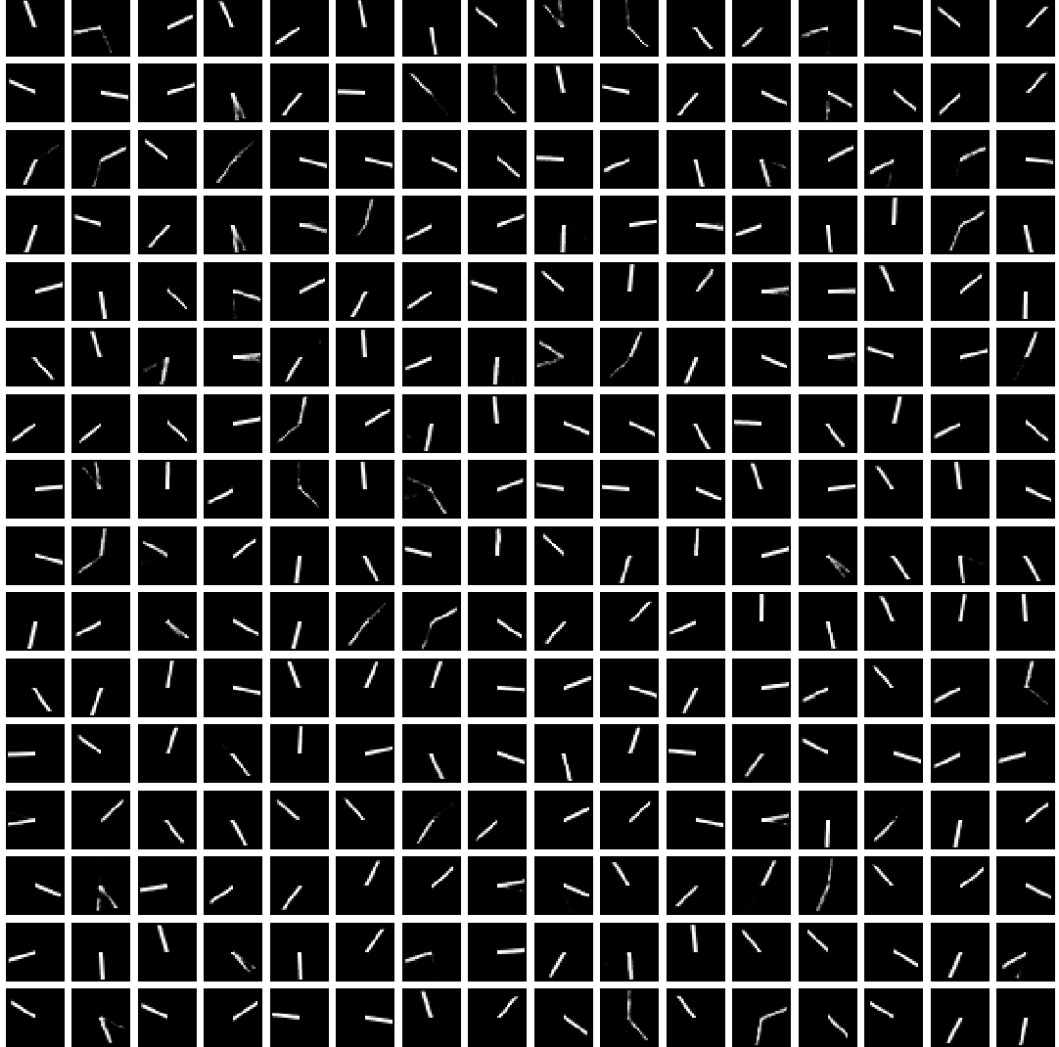

Figure 5: Samples from a VAE trained on the lines dataset described in section 3.1.

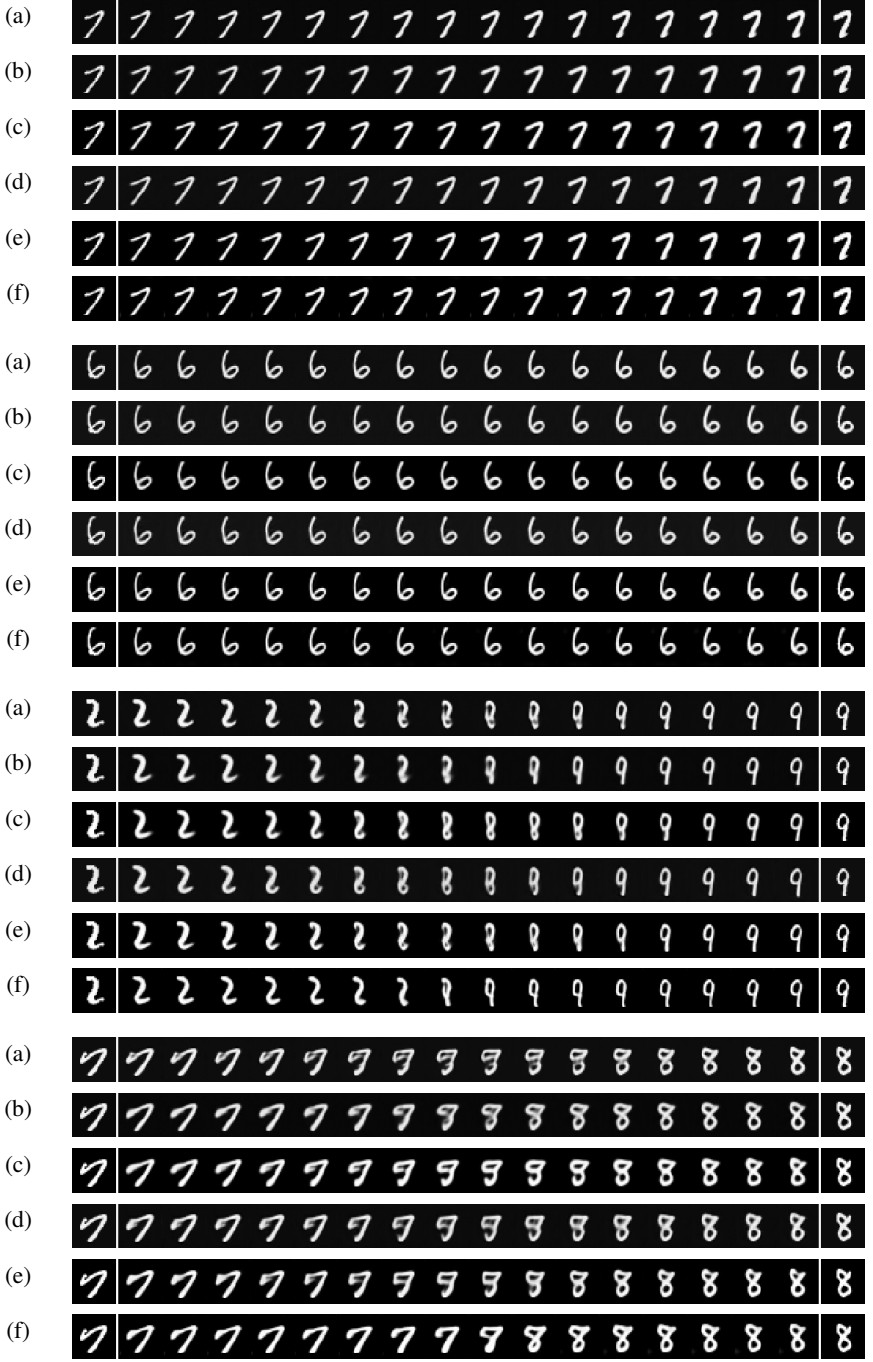

Figure 6: Example interpolations on MNIST with a latent dimensionality of 32 for (a) Baseline, (b) Denoising, (c) VAE, (d) AAE, (e) VQ-VAE, (f) ACAI autoencoders.

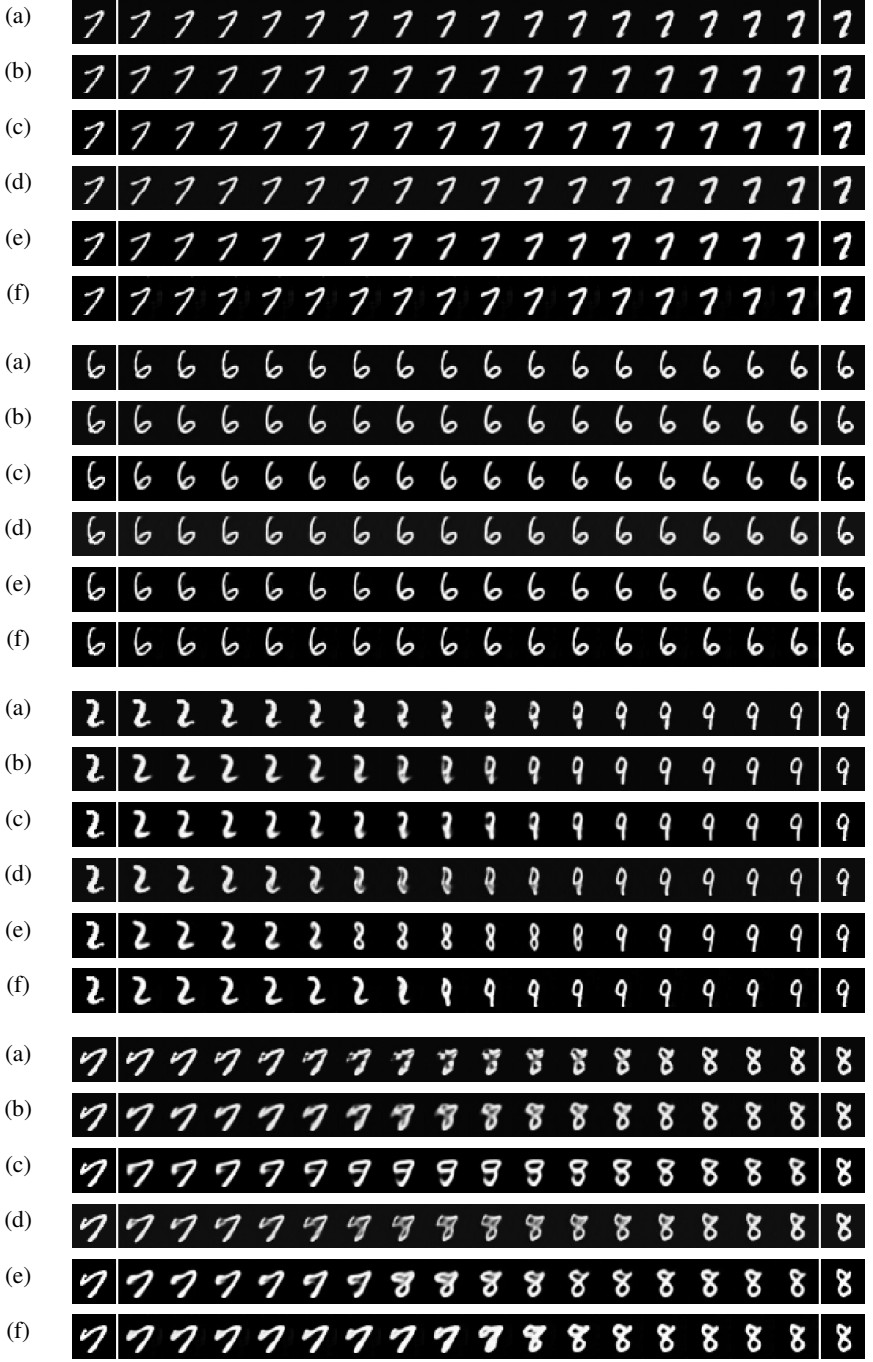

Figure 7: Example interpolations on MNIST with a latent dimensionality of 256 for (a) Baseline, (b) Denoising, (c) VAE, (d) AAE, (e) VQ-VAE, (f) ACAI autoencoders.

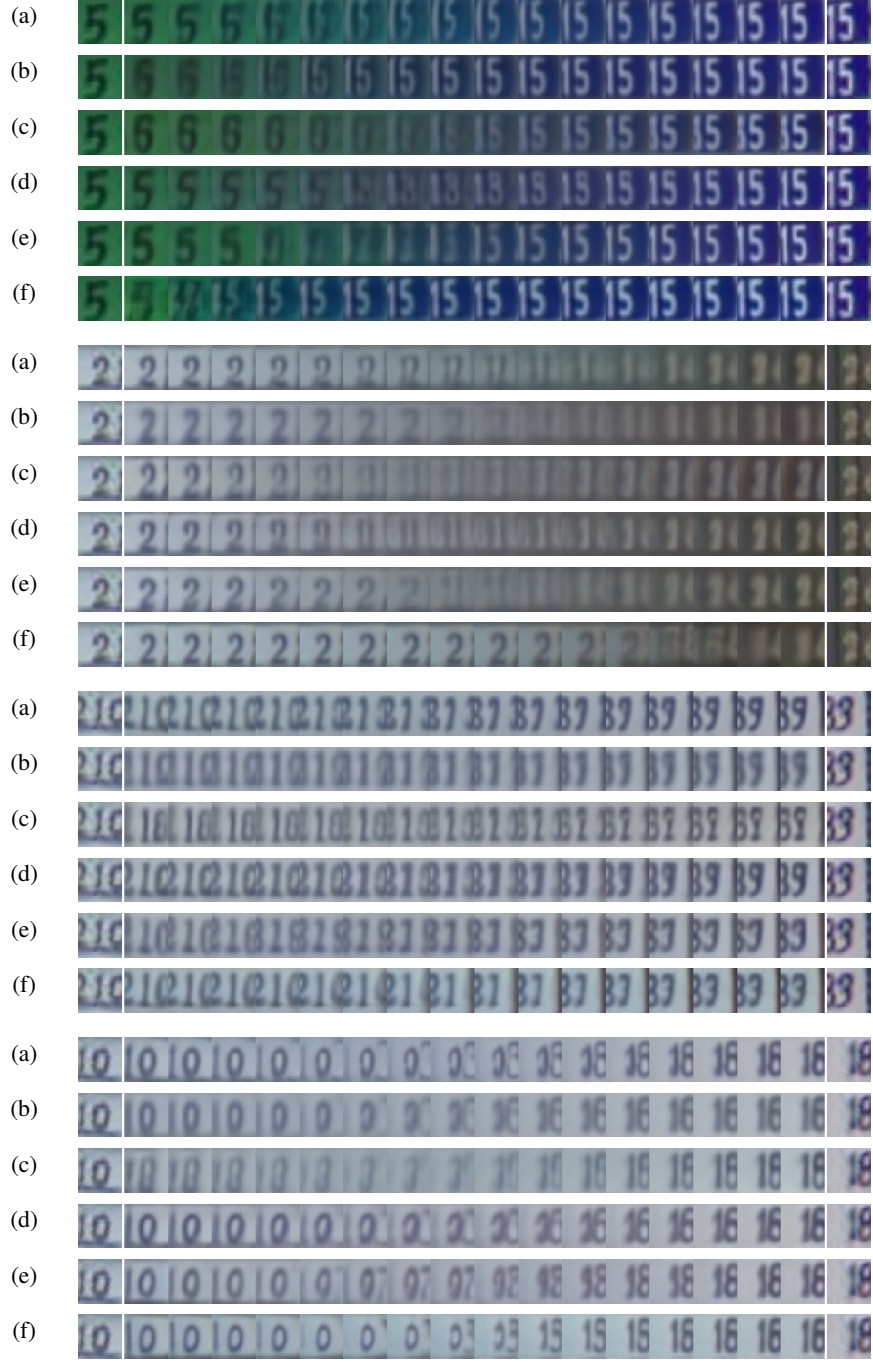

Figure 8: Example interpolations on SVHN with a latent dimensionality of 32 for (a) Baseline, (b) Denoising, (c) VAE, (d) AAE, (e) VQ-VAE, (f) ACAI autoencoders.

(a)

(b)

(c)

(d)

(e)

(f)

(a)

(b)

(c)

(d)

(e)

(f)

(a)

(b)

(c)

(d)

(e)

(f)

(a)

(b)

(c)

(d)

(e)

(f)

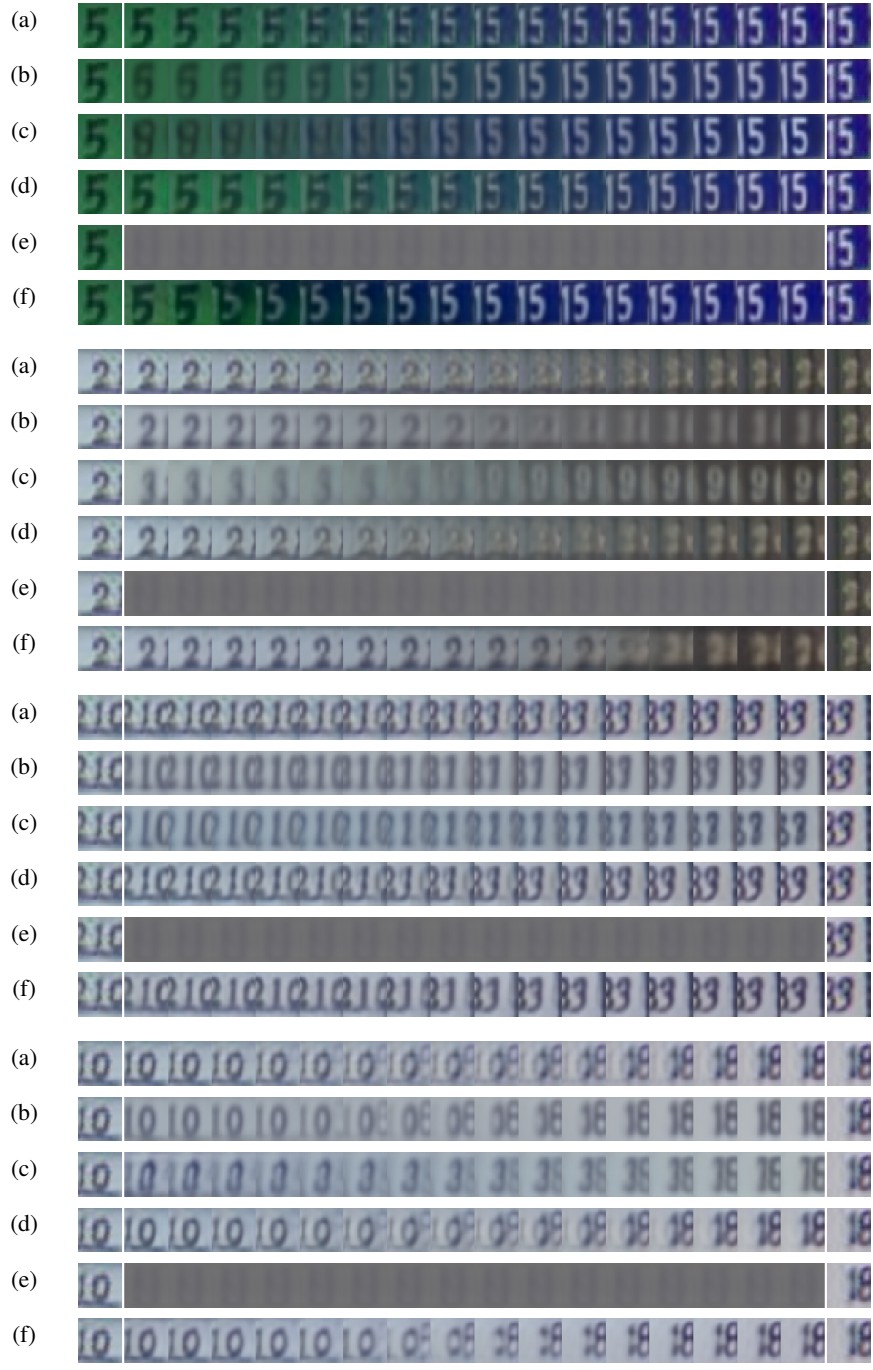

Figure 9: Example interpolations on SVHN with a latent dimensionality of 256 for (a) Baseline, (b) Denoising, (c) VAE, (d) AAE, (e) VQ-VAE, (f) ACAI autoencoders.

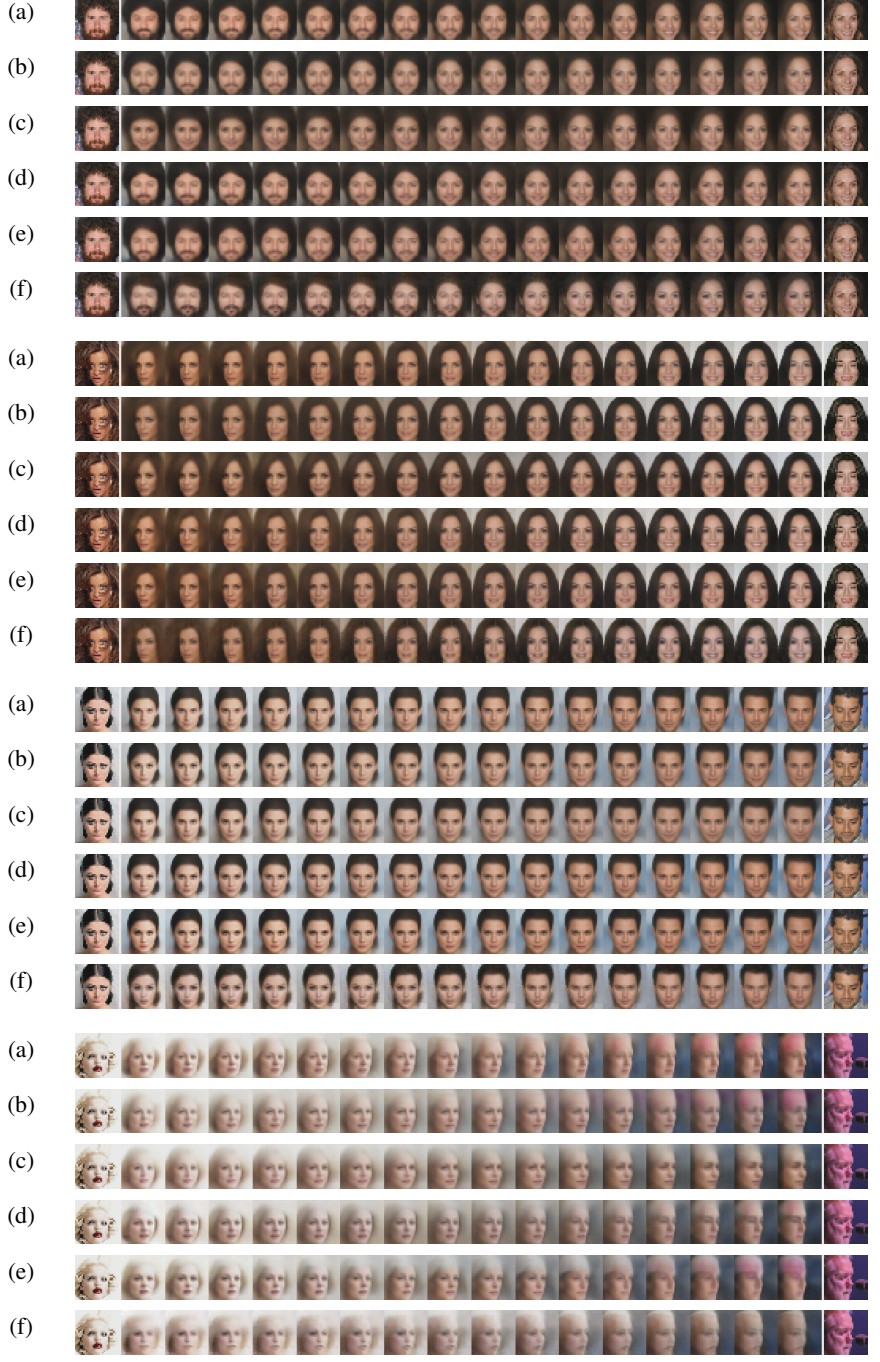

Figure 10: Example interpolations on CelebA with a latent dimensionality of 32 for (a) Baseline, (b) Denoising, (c) VAE, (d) AAE, (e) VQ-VAE, (f) ACAI autoencoders.

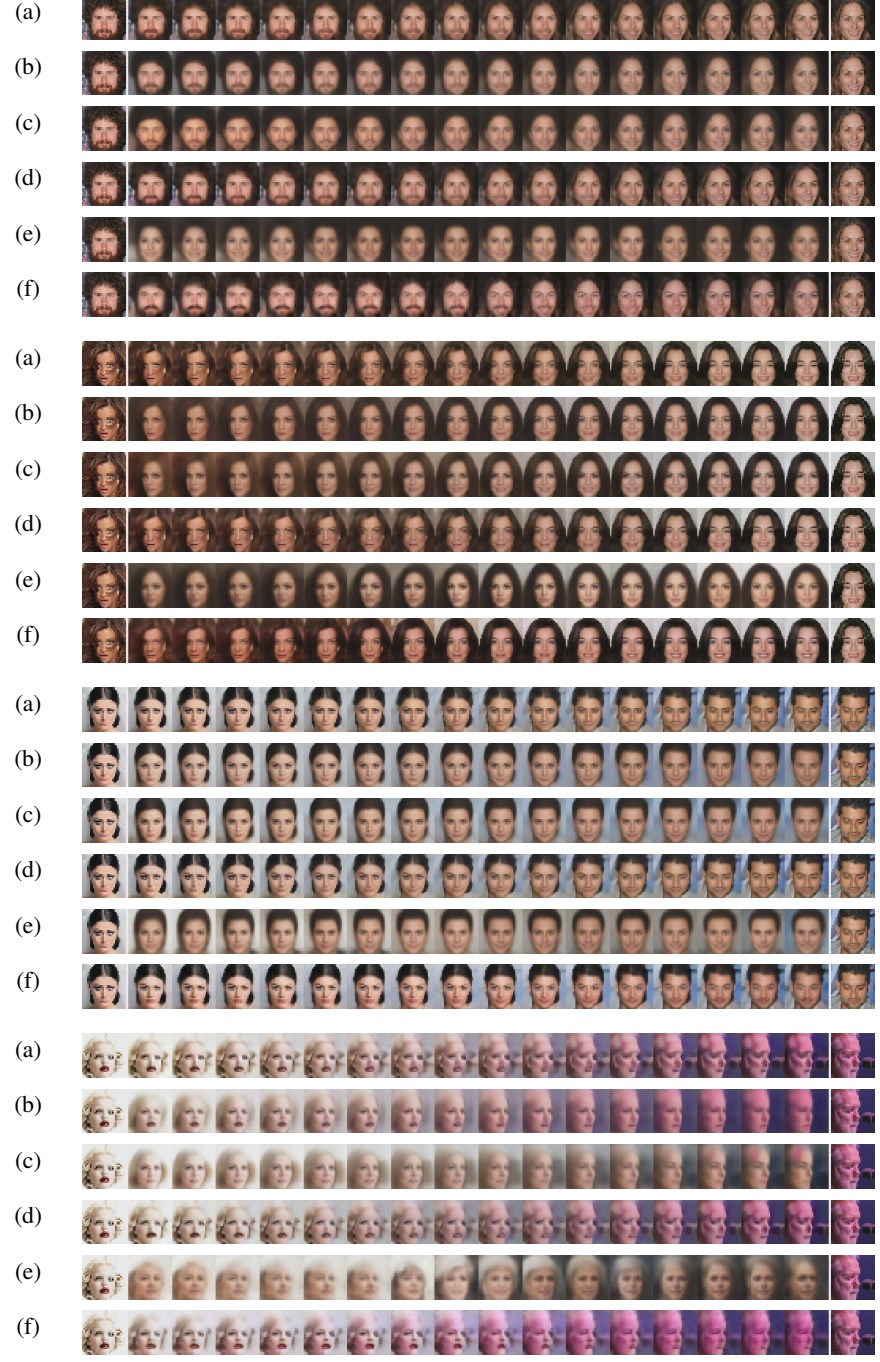

Figure 11: Example interpolations on CelebA with a latent dimensionality of 256 for (a) Baseline, (b) Denoising, (c) VAE, (d) AAE, (e) VQ-VAE, (f) ACAI autoencoders.

