# OpenReview forum: "Understanding and Improving Interpolation in Autoencoders via an Adversarial Regularizer"
_ICLR.cc/2019/Conference_

### Official Review · AnonReviewer2 · 2018-10-31
**An interesting regularized AE algorithm that improves interpolation in latent space**

**Rating:** 9
**Confidence:** 4

**Review:**

This paper proposed an adversarially regularized AE algorithm that improve interpolation in latent space. Specifically, a critic is used to predict the interpolation weight \alpha and encourage the interpolated images to be more realistic. The paper verified the method on a newly proposed synthetic line benchmark and on downstream classification and clustering tasks.

Pros:
1.	A novel algorithm that promotes the interpolation ability of AE
2.	A new synthesized line benchmark to verify the interpolation ability of different AE variants
3.	Strong results on downstream classification and clustering tasks

Cons:
1.	The interplay of the adversarial network (between AE and critic) isn’t very clear and can be improved
2.	Eq. 1, should x be x_1 or a new data other than x1 and x2?
3.	The paper states that the 2nd term of Eq. 1 isn’t crucial. If x is a new data (other than x1 or x2), how can the critic infer \alpha without a reference to x1 or x2?
4.	The paper states that “encouraging this behavior also produce semantically smooth interpolation …”. Besides the empirical evidences from data, it would be better to any some theoretical justifications.

---

> ### Author Response · Authors · 2018-11-14
> **Re: An interesting regularized AE algorithm that improves interpolation in latent space**
>
> Thanks for your review and thoughtful analysis. To address each of your cons in turn:
>
> > The interplay of the adversarial network (between AE and critic) isn’t very clear and can be improved.
>
> The goal of the critic is to predict the interpolation mixing coefficient \alpha; the goal of the autoencoder is to "fool" the critic into outputting \alpha = 0. It can be useful to think of the critic as estimating a divergence between real and interpolated datapoints, and the autoencoder is trying to minimize this divergence. We have added some discussion of this to our paper.
>
> > Eq. 1, should x be x_1 or a new data other than x1 and x2?
>
> It actually can be any real datapoint x - the second term can be computed separately from the first. We have clarified this in our updated draft.
>
> > The paper states that the 2nd term of Eq. 1 isn’t crucial. If x is a new data (other than x1 or x2), how can the critic infer \alpha without a reference to x1 or x2?
>
> The critic must infer \alpha from common artifacts of interpolated datapoints alone. This is best illustrated in Figure 3(a) - note that as the interpolation morphs from one endpoint to the other, the image becomes dimmer and closer to a "dot" in the middle of the image. In this case, it is easy to infer \alpha based on the length and brightness of the line. This is exactly the kind of behavior that ACAI seeks to discourage, and we find it's effective in practice. We have added some additional discussion of this point to our paper.
>
> > The paper states that “encouraging this behavior also produce semantically smooth interpolation …”. Besides the empirical evidences from data, it would be better to any some theoretical justifications.
>
> Our approach can be viewed in the framework of adversarial divergences, where the critic network is being used to estimate a divergence . Of course, the exact form of this divergence is not clear, but it does provide a connection to the GAN theory literature. We have made this connection explicit in our updated draft.

---

### Official Review · AnonReviewer1 · 2018-11-02
**Review for "Understanding and Improving Interpolation in AE via an Adversarial Regularizer - Interesting Paper with good results.**

**Rating:** 8
**Confidence:** 3

**Review:**

Summary: The authors propose a new approach to encourage valid interpolation in Auto-Encoders (AE). It is based on a regularization procedure involving a critic network judging the realistic nature of reconstructed data point from its mixed latent representations by recovering the mixing coefficient. The authors show that this approach does indeed improve the quality of interpolated samples on few tasks. A synthetic tasks of lines interpolation (proposing new Mean Distance and Smoothness metric for this task), classification task (with a single-layer classifier) from the latent space representation and finally a clustering accuracy on the latent space. On the proposed regularization method seems to help significantly compared to commonly used AE architectures (Basic AE, Denoising AE, Variational AE, Adversarial AE and VQ-VAE).

This paper was a very interesting read, and the work seems to be of significance for the unsupervised learning community.
It was clearly written and conveys the contributions clearly and the experimental results and their interpretations seem valid.

The proposed approach of a critic based regularizer is a simple but seemingly important addition that contributes to improving interpolation in AE significantly and even show impact "downstream tasks" as the authors put it.

Few comments/questions come to mind:

- For the critic Loss L_d in equation (1) , the authors mention that the \gamma based second term (that should ensure that the critic outputs 0 for non-interpolated inputs and expose the critic to realistic data even if the AE reconstruction is poor)  does not seem to be crucial in your approach but stabilized the adversarial training. Could you somehow quantify this. It seems like stability of the adversarial training should be paramount to your method to make sure the AE learns a better latent representation. This comment, even though I assume it well-founded, seems a bit of a contradiction.

- For the Lines synthetic data. It was chosen to use a 32x32 image size with 16 points length lines. This configuration does quantize directly the angles your measures can distinguish. Below a certain angle differences (or delta), 2 angles must have the same pixel representation, i.e. exact overlapping lines. My question is simple: What is the smallest angle you can use/distinguish or, how many exact unique lines can you have?

Overall this is a good paper that deserves publications.

---

> ### Author Response · Authors · 2018-11-14
> **Re: Review for "Understanding and Improving Interpolation in AE via an Adversarial Regularizer - Interesting Paper with good results.**
>
> Thanks for your review, we are glad you found the paper interesting and significant. To address your questions and comments:
>
> > For the critic Loss L_d in equation (1) , the authors mention that the \gamma based second term (that should ensure that the critic outputs 0 for non-interpolated inputs and expose the critic to realistic data even if the AE reconstruction is poor)  does not seem to be crucial in your approach but stabilized the adversarial training. Could you somehow quantify this. It seems like stability of the adversarial training should be paramount to your method to make sure the AE learns a better latent representation. This comment, even though I assume it well-founded, seems a bit of a contradiction.
>
> We agree that this comment should be expanded on, and we have done so in our updated draft. To clarify, when we say it "helped stabilize the adversarial learning process", we mean that a) it allowed us to use the same value of \lambda across all of our experiments and still achieve good results and b) it resulted in smooth convergence of the autoencoder's loss. We note that stability of the adversarial learning process was not an issue in general, in the sense that stability across runs was not an issue and our model never "collapsed" to a bad solution.
>
> > For the Lines synthetic data. It was chosen to use a 32x32 image size with 16 points length lines. This configuration does quantize directly the angles your measures can distinguish. Below a certain angle differences (or delta), 2 angles must have the same pixel representation, i.e. exact overlapping lines. My question is simple: What is the smallest angle you can use/distinguish or, how many exact unique lines can you have?
>
> Our code for synthesizing line images uses anti-aliasing, so for example a line with angle 0.3 and another with angle 0.300001 will be rendered differently. As a result, the number of unique lines is actually up to floating point precision. We think some confusion about this probably stems from the fact that we referred to the line images as "black-and-white"; we have updated the language in the paper to say "grayscale".

---

> > ### Comment · AnonReviewer1 · 2018-12-03
> > **Thanks and good luck**
> >
> > I would like to thank the authors for addressing my feedback. This comforts me in the rating I gave to their paper.
> > Good luck.

---

### Official Review · AnonReviewer3 · 2018-11-02
**Regularize interpolation or regularize manifold?**

**Rating:** 7
**Confidence:** 4

**Review:**

Main idea:
This paper investigates the desiderata for a successful interpolation:
1) Interpolation looks realistic;
2) The interpolation path is semantically smooth.
An adversarial regularizer is proposed to achieve 1), and in practice 2) may also satisfied.
To evaluate the method, they introduce a synthetic dataset with line images and compare with different autoencoder methods without the interpolation regularization.
For real data, they show that the interpolation regularized autoencoder (i.e. ACAI) leads to a better unsupervised representation.

Questions:
1. Do we really need every interpolated point to be realistic (i.e. similar to a data point in the train-set)? I believe that there exists an interpolation between two totally different objects can never be observed.
2. Do we need interpolation points to form a semantically smooth morphing? I guess this is a desired property for continuous generators, but it seems not necessary in general.
3. The gamma in the 2nd term in (1) is confusing. If gamma = 1, I understand it forces to predict alpha = 0 since x is real. But if gamma < 1, the average in data space may be very blurry thus not realistic at all. How does gamma affect the optimization?
4. ACAI looks very similar to LSGAN: by giving "0" label to real data and "alpha" label to fake data; in LSGAN, alpha = 1.
Have you tested a LSGAN like regularizer?
5. The baselines are not representative: since ACAI introduces an adversarial regularizer, you should compare with other GAN techniques induced regularizers, such as WGAN regularized autoencoder.

After rebuttal:
See the long discussion below. I tend to believe that a good interpolation is not only a way to do sanity check but also a nice property to explicitly control in representation learning.

---

> ### Author Response · Authors · 2018-11-14
> **Re: Regularize interpolation or regularize manifold?**
>
> Thanks for your thorough review and questions. We've answered your questions below and have updated our draft to clarify.
>
> > Do we really need every interpolated point to be realistic (i.e. similar to a data point in the train-set)? I believe that there exists an interpolation between two totally different objects can never be observed.
>
> We are interested in latent spaces where interpolations produce realistic outputs across the entirety of the interpolation because this suggests some form of continuity in the latent space (as illustrated in FIgure 1). Our paper asks whether this property also results in an improved representation for downstream tasks. If an intermediate point was not realistic, the latent space might not have this property.
> Thanks for pointing out that in some cases it's not obvious that there is a smooth and realistic path between two datapoints. We think two good examples of this are in Figure 6, bottom, where we interpolate between different MNIST digits. We find that even though there is no real digit which is at the midpoint of, for example, a 2 and a 9, the midpoint of ACAI's interpolation still appears realistic. We have added a note about this to our paper.
>
> > Do we need interpolation points to form a semantically smooth morphing? I guess this is a desired property for continuous generators, but it seems not necessary in general.
>
> We agree that smoothness is not required for high-quality learned features -- for example, the Denoising Autoencoder fared well on our classification experiments despite producing poor interpolations. However, we are interested in the opposite, namely whether the ability to perform latent-space manipulations like interpolation suggest a better learned representation. We have added some clarification of this point in our paper.
>
> > The gamma in the 2nd term in (1) is confusing. If gamma = 1, I understand it forces to predict alpha = 0 since x is real. But if gamma < 1, the average in data space may be very blurry thus not realistic at all. How does gamma affect the optimization?
>
> Note \hat{x} is a reconstruction of x, so in practice \gamma*x + (1 − \gamma)*\hat{x} will be quite similar to x as long as \hat{x} is a reasonable reconstruction. In other words, we are not interpolating between two totally different points, so typically the blurriness you might expect from pixel-space mixing won't be present. We have added some additional discussion of gamma and this term to our paper.
>
> > ACAI looks very similar to LSGAN: by giving "0" label to real data and "alpha" label to fake data; in LSGAN, alpha = 1. Have you tested a LSGAN like regularizer?
>
> You're right that the LSGAN loss function and our regularization term are similar in the sense that both measure a squared error between the critic's output and a scalar. The difference is that the LSGAN is designed for use on a GAN-based generative model; our regularizer is designed as a regularizer for an autoencoder. As a result, the scalar in the LSGAN objective is a fixed hyperparameter whereas we regress the interpolation amount \alpha. We added some discussion of the LSGAN objective to our paper.
>
> > The baselines are not representative: since ACAI introduces an adversarial regularizer, you should compare with other GAN techniques induced regularizers, such as WGAN regularized autoencoder.
>
> Note that the Wasserstein Autoencoder (WAE) is actually equivalent to an adversarial autoencoder when using a GAN loss; in the WAE paper [1] they write "When c is the squared cost and D_Z is the GAN objective, WAE coincides with adversarial auto-encoders". Our paper includes the adversarial autoencoder as a baseline (labeled AAE in tables and described in Section 3.2, paragraph 4). We added a citation to [1] to clarify this.
>
> [1] Ilya Tolstikhin, Olivier Bousquet, Sylvain Gelly and Bernhard Schoelkopf. "Wasserstein Auto-Encoders", ICLR 2017.

---

> > ### Comment · AnonReviewer3 · 2018-12-04
> > **This is a hidden discussion after rebuttal**
> >
> > Sorry that I didn't realize the discussion between the authors and me was private! I replied to AC's question which was private making everything private afterwards.
> > I think it is worth an open discussion by more people. So I post the discussion here.
> >
> > *** By reviewer 3 ***
> > This is an interesting idea, but I'm still not sure its practicality for autoencoders. I will rephrase and elaborate my concerns:
> >
> > > R3: "The baselines are not representative; you should compare with other GAN techniques induced regularizers, such as WGAN regularized autoencoder."
> > A: "WAE = AAE; Our paper includes the adversarial autoencoder as a baseline."
> >
> > I'm sorry the question was not clear. In fact, I meant to compare other GAN regularizers for the output of the decoder (AAE regularizes the code), which is quite common due to the popularity of CycleGAN, and it indeed improves significantly the quality of the outputs.
> > As I asked: should we "regularize the interpolation or regularize the image of the decoder"?
> > I think the latter is the main desideratum for autoencoders.
> > Interpolation regularizer is one way to achieve that; and the proposed ACAI in my opinion is a generalized LSGAN regularizer (may be the motivations are different).  But since there is no comparison between that and ACAI, I'm not sure if this interpolation extension plays an important role.
> >
> > > Regarding the philosophy of interpolation:
> > 1) Interpolation looks realistic;
> > 2) The interpolation path is semantically smooth,
> >
> > I am not sure if there is a clear connection between a good interpolation and a good representation learning, since there are good discrete representation learning and as the authors mentioned the denoising AE could perform better despite producing bad interpolations.
> > More experiments are needed to gain a deeper understanding. The evaluation of interpolation on a toy dataset is far from satisfactory.
> >
> >
> > *** By authors ***
> > R3, thanks for clarifying your initial comments and for your additional discussion. We think there remains some misunderstanding about the scope and claims of our paper.
> >
> > 1) Our paper focuses on autoencoders and representation learning, not generative models. GANs are generative models, and ACAI is a regularizer for autoencoders. While ACAI includes an adversarial training process and a "critic", it otherwise has very little in common with GANs and the resulting autoencoder is not a generative model. Similarly, while the loss function has some similarities with the LSGAN loss function (i.e., they both use a least-squared error loss), it has very little in common with an LSGAN because an LSGAN is a generative model and not an autoencoder or a technique for learning representations. We agree that it would be interesting to study the effect of regularizing the decoder of an autoencoder in similar ways to the generator in a GAN, but this is outside the scope of our paper. More specifically, GANs are not representation learning techniques, they are generative models; so, there is no way to test their representation learning capabilities (as is the focus of our paper).
> >
> > 2) We do not claim that "good interpolation implies a good representation and a good representation implies good interpolation". In contrast, we ask "given an autoencoder which reconstructs well but interpolations poorly (our Baseline), can we improve the quality of its interpolations, and does improving the interpolation quality improve the representation learned?" Note that the first is a claim of causality, ours is a test of an intervention. As an aside, we are not the first to study or point out this potential connection; see e.g. "Better Mixing via Deep Representations" by Bengio et al.
> >
> > Based on this discussion, we have included some additional statements in our updated draft to make it clear what the scope and claims of our paper are. We hope this clarifies the intention of our paper.
> >
> >
> > *** By authors ***
> > R3, we believe have addressed your concerns and clarified some of your points. Do you have an updated impression of our paper? Thanks for your consideration.
> >
> >
> > *** By reviewer 3: EXPERIMENT REQUEST ***
> > I like your question: "given an autoencoder which reconstructs well but interpolations poorly (our Baseline), can we improve the quality of its interpolations, and does improving the interpolation quality improve the representation learned?"
> > This should be added to the paper with an emphasis.
> >
> > My main concern still remains: is the good representation coming from a GAN regularized autoencoder (since your interpolation formulation is very similar to that of LSGAN) or because of the improved interpolation (then it's your contribution)?
> > I found the experiments insufficient unless you compared with such a baseline (e.g. LSGAN regularized autoencoder) on representation learning.

---

> > > ### Author Response · Authors · 2018-12-05
> > > **Re: This is a hidden discussion after rebuttal**
> > >
> > > R3, thank you for noticing the comments were not public and making the discussion public.
> > >
> > > > I like your question: "given an autoencoder which reconstructs well but interpolations poorly (our Baseline), can we improve the quality of its interpolations, and does improving the interpolation quality improve the representation learned?"
> > > > This should be added to the paper with an emphasis.
> > >
> > > We are glad this question clarified your understanding of the paper. Unfortunately, the time period for us to be able to make revisions to the paper is over, so we can't update the PDF. However, we can assure you we will include and emphasize (e.g. boldface) this text in an updated draft.
> > >
> > > > My main concern still remains: is the good representation coming from a GAN regularized autoencoder (since your interpolation formulation is very similar to that of LSGAN) or because of the improved interpolation (then it's your contribution)?
> > > > I found the experiments insufficient unless you compared with such a baseline (e.g. LSGAN regularized autoencoder) on representation learning.
> > >
> > > Can you describe in more detail what you mean by an LSGAN regularized autoencoder? Our model is quite different from a GAN, since it is an autoencoder and not a generative model (there is no way to draw samples from it). While it uses a critic and an adversarial learning process, it otherwise has very little in common with GANs. If you mean an autoencoder whose latent space is regularized by a critic, I think that baseline is represented by our inclusion of an AAE. If you have a specific model architecture or loss function in mind, we would be happy to include it in our experiments.

---

> > > > ### Comment · AnonReviewer3 · 2018-12-05
> > > > **Re: Can you describe in more detail what you mean by an LSGAN regularized autoencoder?**
> > > >
> > > > To improve AE by GAN is quite common due to CycleGAN (Zhu et al. 2017, Unpaired Image-to-Image Translation using Cycle-Consistent Adversarial Networks) and pix2pix (Isola et al. 2016, Image-to-Image Translation with Conditional Adversarial Nets). A LSGAN regularized AE is almost equivalent to a CycleGAN except that you only do a half cycle here:
> > > >
> > > > Given a AE: x -> z -> \hat{x} with parameterization \hat{x} = G(z), z = F(x). The critic is trained by minimizing
> > > > L_critic = (D(x) - 0)^2 + (D(\hat{x}) - 1)^2
> > > > and the AE is trained by minimizing
> > > > L_AE = || x - \hat{x} || + lambda * (D(\hat{x}) - 0)^2
> > > >
> > > > This is fairly similar to your objective function in my opinion. So I was asking for a comparison.

---

> > > > > ### Author Response · Authors · 2018-12-05
> > > > > **Re: LSGAN regularized autoencoder**
> > > > >
> > > > > Thanks very much for your clarifications. We now understand what you were describing as a baseline. A few comments on this -
> > > > >
> > > > > 1) We will implement this approach and update this comment thread with the results. Unfortunately, we cannot update the paper draft anymore during the review period, so we will have to just copy results here and update the paper in subsequent drafts. We will also push the code for this approach to our anonymous repository so that you can verify that we are implementing what you've described.
> > > > >
> > > > > 2) The difference between what you are describing and what we propose is that our critic only ever sees reconstructions and interpolants - it never sees real points. In what you described, we understand the goal to be to make reconstructions more realistic. We instead enforce that interpolants look like reconstructions, which we could expect to have a very different impact. Our paper is focused on improving interpolation quality rather than reconstruction quality - we do not expect our approach to improve reconstruction quality (compared to a baseline without the regularizer).
> > > > >
> > > > > 3) We want to point out a distinction between pix2pix/cycleGAN and autoencoders. For completeness, we first define our understanding of pix2pix, CycleGAN, and an autoencoder below.
> > > > > - pix2pix consists of a generator which maps an input x to an output \hat{y} = G(x). The discriminator tries to distinguish between pairs of (x, \hat{y}) (generated pair) and (x, y) (real pairs).
> > > > > - CycleGAN contains two generators, one to map from x -> y and one for y -> x. Call the first one G(x) and the second F(y). Two discriminators D_x and D_y are trained to distinguish between outputs of F(x) vs. real x's and outputs of G(y) and real y's. The CycleGAN loss enforces that F(G(x)) = x and G(F(y)) = y and that G and F fool D_y and D_x respectively.
> > > > > - An autoencoder uses an encoder to map x to a latent z, and then from z back to the latent space x. It's typically trained to reconstruct x accurately. The latent z can be used for representation learning and semantic manipulation of data (such as interpolation). We introduce a regularizer which also encourages interpolants to appear similar to reconstructions.
> > > > > We want to point out that neither pix2pix nor CycleGAN contain a latent code or an encoder/decoder, so we don't think of them as autoencoders. While CycleGAN does include a loss which encourages cycle-consistency, there is no latent code, and so there is no opportunity for interpolation or representation learning. We believe the primary similarity between CycleGAN and ACAI is that both use a discriminator to learn and minimize a divergence between implicit distributions, but to us this is a commonality to any model using a critic. We have some discussion of this in section 2.1 of our current draft, but we can expand this discussion to include a comparison to CycleGAN and pix2pix in future drafts.

---

> > > > > > ### Author Response · Authors · 2018-12-06
> > > > > > **LSGAN regularized autoencoder results, part 1**
> > > > > >
> > > > > > We have implemented the LSGAN regularized AE as you described and have (anonymously) pushed the code to https://github.com/anonymous-iclr-2019/acai-iclr-2019/blob/master/lrae.py
> > > > > > The loss is implemented here: https://github.com/anonymous-iclr-2019/acai-iclr-2019/blob/master/lrae.py#L61
> > > > > >
> > > > > > We have run this autoencoder on the lines dataset. We tried lambda in {0.01, 0.02, 0.05, 0.1, 0.2, 0.5, 1.0}. The best setting of lambda achieved a Mean Distance of 3.62e-3 and a Smoothness of 0.51. For high settings of lambda, the autoencoder collapses to producing a single output. For comparison, the baseline autoencoder (equivalent to setting lambda = 0) achieved Mean Distance of 6.88e-3 and a Smoothness of 0.44 (lower is better for both). It appears qualitatively and quantitatively that (on this task) including this additional loss term improves reconstruction quality (lowering the Mean Distance) but makes the interpolation quality slightly worse (lowering the Smoothness). The interpolations exhibit sudden jumps (similar to the VAE), hence the poor smoothness score. This follows our intuition - the regularizer you suggested will make reconstructions  closer to real data (i.e., more realistic) but doesn't have a mechanism to improve interpolations or change the structure of the latent space. For comparison, the ACAI regularized autoencoder achieves a Mean Distance 0.24e-3 and a Smoothness of 0.10.
> > > > > >
> > > > > > We will now run the autoencoder on our representation learning experiments on real datasets and will report back with results.

---

> > > > > > > ### Author Response · Authors · 2018-12-07
> > > > > > > **LSGAN regularized autoencoder results, part 2**
> > > > > > >
> > > > > > > We have now tested an autoencoder using this regularizer on our representation learning experiments. As a reminder, we first train an autoencoder on MNIST, SVHN, and CIFAR-10. We then use the latent codes as a learned representation for a single-layer classifier and report the accuracy of the classifier on the test set. Denoting the LSGAN Regularized Autoencoder as "LRAE", we obtained the following results (with Baseline and ACAI results included for reference):
> > > > > > >
> > > > > > >                                      | Baseline | ACAI  | LRAE
> > > > > > > MNIST, d_z = 32          | 94.90      | 98.25 | 95.66
> > > > > > > MNIST, d_z = 256        | 93.94      | 99.00 | 96.94
> > > > > > > SVHN, d_z = 32           | 26.21      | 34.47 | 22.49
> > > > > > > SVHN, d_z = 256         | 22.74      | 85.14 | 30.77
> > > > > > > CIFAR-10, d_z = 256   | 47.92      | 52.77 | 47.99
> > > > > > > CIFAR-10, d_z = 1024 | 51.62      | 63.99 | 50.26
> > > > > > >
> > > > > > > The LRAE improves over the baseline in some cases, but not consistently. Since the additional loss term/critic in LRAE is satisfied by making reconstructions more realistic, we hypothesize that it does not change the structure of the latent space. This would explain why it does not generally improve representation learning performance. In contrast, ACAI has the specific goal of modifying the structure of the latent space by making interpolants appear more like reconstructions. This results in improved representation learning performance.
> > > > > > >
> > > > > > > We believe that these additional experiments further strengthen our claim that improving interpolation behavior can also produced a better learned representation. We also believe this addresses your main concern:
> > > > > > > > My main concern still remains: is the good representation coming from a GAN regularized autoencoder (since your interpolation formulation is very similar to that of LSGAN) or because of the improved interpolation (then it's your contribution)?
> > > > > > > The results of these experiments definitively show that the improved performance comes from the specific form and goal of ACAI and not simply from that our approach uses a critic. We hope this convinces you of the merit of our submission.

---

> > > > > > > > ### Comment · AnonReviewer3 · 2018-12-07
> > > > > > > > **Interesting results**
> > > > > > > >
> > > > > > > > I appreciate your quick experiments for addressing my concerns!
> > > > > > > >
> > > > > > > > Now I'm convinced ACAI is a quite interesting method:
> > > > > > > > It seems very important for the critic to see reconstructions and interpolants only. I tend to believe this somehow smooth the latent space, while LRAE doesn't make a full use of the encoder since by increasing d_z the performance only increases marginally.
> > > > > > > >
> > > > > > > > I believe ACAI deserves more visibility to our community.

---

> > > > > > > > > ### Author Response · Authors · 2018-12-07
> > > > > > > > > **Thanks!**
> > > > > > > > >
> > > > > > > > > Thanks for engaging in discussion with us, suggesting additional experiments, and being open to updating your review.

---

### Author Response · Authors · 2018-11-14
**Updated draft**

Thanks to all of the reviewers for their feedback on our paper. We have addressed each reviewer's comments individually and have also uploaded an updated draft based on the suggestions. The changes include the following:
- Clarified why smooth and realistic interpolations may potentially lead to better reconstructions in the introduction
- Framed our objective as minimizing an adversarial divergence between reconstructions and interpolants
- Clarified the second term of the critic loss involving \gamma and gave additional justification for this term
- Added comparison of the ACAI and LSGAN critic losses
- Gave additional intuition as to how the critic could potentially regress \alpha when only being shown a single image at a time
- Referred to our lines images as "greyscale" rather than "black and white"
- Noted that the AAE baseline we included has also subsequently been referred to as Wasserstein Autoencoder
- Pointed out some cases where interpolations can be smooth and realistic despite interpolating between dissimilar points

We hope these changes address any concerns the reviewers have.

---

### Public Comment · (anonymous) · 2018-11-14
**Does the technique help in getting better random samples?**

For example, if the proposed regularizer is applied to a VAE, does it help in getting better random samples by decoding z ~ N(0, 1)?

---

> ### Author Response · Authors · 2018-11-14
> **Re: Does the techinique help in getting better random samples?**
>
> Thanks for your question. In general we do not expect this regularizer to improve the sample quality of a given autoencoder, since the critic's primary objective is to discriminate between interpolants and reconstructions (not interpolants and "real" data). The goal instead is to take an autoencoder which already reconstructs well but interpolates poorly and improve the quality of the interpolations. The VAE typically has the opposite problem - it reconstructs poorly but interpolates smoothly. In other words, the latent space of the VAE is already "continuous" in some sense (due to the enforcement of the prior) but many regions in latent space map to "unrealistic" (i.e. blurry) outputs. So, we aren't sure whether our regularizer would improve VAE reconstructions. It would be pretty straightforward to try using our publicly-available code, though!

---

### Meta-Review · Area_Chair1 · 2018-12-14
**Strong paper**

**Confidence:** 4
**Recommendation:** Accept (Poster)

**Metareview:**

The reviewers have reached a consensus that this paper is very interesting and add insights into interpolation in autoencoders.